# Generalized Inference Time Unlearning — Effective for a Fraction of the Cost

## Abstract

Large Language Models (LLMs) can memorize and regurgitate sensitive training data, creating significant privacy and safety risks. While existing unlearning aim to address these risks, current methods are often computationally prohibitive and/or significantly degrade model utility. We introduce a framework for Inference-Time Unlearning, a new paradigm that steers an LLM's output at inference time using small secondary models, without altering the base model's weights. Through extensive experiments with LLMs we demonstrate that our method is highly effective at removing targeted verbatim and semantic knowledge, is orders of magnitude more computationally efficient—through profiling of more than 1,200 models—than traditional approaches, and fully preserves the base model's general capabilities. We then explore efficacy in unlearning visual semantics in generative image models and find similar evidence of effectiveness. Collectively, the framework offers a practical, scalable, and low-cost solution for selective forgetting, enabling more responsible and adaptable model deployment. All code to reproduce this work is available at the following anonymous link.

## 1 Introduction

Large Language Models (LLMs) have demonstrated remarkable capabilities, achieving state-of-the-art performance on a diverse array of natural language tasks and becoming integral to a wide range of applications (Brown et al., 2020; Touvron et al., 2023; DeepSeek-AI, 2024). However, the very scale that enables their powerful generalization also creates significant challenges (Weidinger et al., 2021). LLMs have been shown to memorize and regurgitate portions of their training data, including personally identifiable information (PII), proprietary text, and harmful content (Carlini et al., 2021). This behavior creates urgent privacy, safety, and copyright concerns (Henderson et al., 2023), conflicting with principles like the "right to be forgotten" mandated by regulations such as the GDPR (Voigt & dem Bussche, 2017).

The most straightforward solution to remove unwanted data from an LLM is to retrain it from scratch on a sanitized dataset (Bourtoule et al., 2021). Given that training a flagship model requires vast computational resources, this approach is economically and practically infeasible for frequent unlearning requests. For instance, training Meta's Llama 3 70B model consumed approximately 1.6 million GPU-hours, and other state-of-the-art models demand similarly massive-scale resources (Hoffmann et al., 2022; Grattafiori et al., 2024; DeepSeek-AI, 2025). Consequently, the field of machine unlearning has emerged to develop methods that can efficiently remove data's influence from a trained model (Nguyen et al., 2025). Prevailing techniques often rely on fine-tuning the full model, using methods like gradient ascent to maximize the likelihood of forgetting specific data or negative preference optimization to steer the model away from undesired outputs (Eldan & Russinovich, 2023; Jang et al., 2023). While less expensive than complete retraining, these methods still require costly gradient updates on the entire large model and can often lead to a degradation of the model's overall capabilities, a phenomenon known as catastrophic forgetting (Kirkpatrick et al., 2017).

In this work, we propose a new paradigm inspired by product of experts (Hinton, 1999) and speculative decoding (Leviathan et al., 2023): **Inference-Time Unlearning**, which simulates the effects of unlearning in a model's outputs without modifying its parameters. Our method, Divergence Decoding (DD), requires no modifications to the weights of the large base model. Instead, it guides

text generation at inference-time by using a pair of much smaller, specialized models. One small model is fine-tuned on the data to be forgotten (the "forget set"), while another is tuned on a proxy for the data to be retained. By modifying the logits of the base model with the difference of the "retain" and "forget" models, our method steers the output distribution away from unwanted content while leaving general knowledge and model utility largely unaffected, so that the model behaves as if the content had been unlearned. This method is applicable to API-locked models or unlearning use cases (e.g., financial research) that do not require the forget set of data to be protected.

Our paper makes three primary contributions to the literature on machine unlearning:

1. **Efficacy**: We demonstrate that Inference-Time Unlearning effectively suppresses both verbatim and semantic recall from the forget set in the model's outputs, closely matching the behavior of a retrained model on the standard unlearning benchmarks MUSE and TOFU. Further, we apply our method to VQGAN image generation models (Esser et al., 2021) and find some evidence of unlearning visual semantics.

2. **Utility Preservation**: Our method maintains the model's performance on general knowledge and standard evaluation benchmarks. Because the base model's weights remain unchanged, the impact on its core capabilities is minimal, outperforming prior methods in preserving utility as the number of unlearning requests grows.

3. **Efficiency**: By restricting fine-tuning to small models (with orders of magnitude fewer parameters than the base LLM), our approach has drastically reduced the computational cost compared to true unlearning. For example, we find that even simple tri-gram based LMs are effective. This makes on-demand unlearning practical and scalable.

We show that our approach provides a practical, low-cost, and effective solution to the critical problem of selectively forgetting information in LLMs, paving the way for more responsible and adaptable deployment of these powerful models.

## 2 RELATED LITERATURE

**Removing knowledge from model weights.** Model providers use methods such as Supervised Safety Fine-tuning and RLHF to finetune their models to reduce the likelihood of generating certain content when aligning the models (Touvron et al., 2023; Achiam et al., 2024). For post-alignment methods, a variety of different variations of finetuning aim to remove knowledge from the model's weights while damaging its utility as little as possible. (Jang et al., 2023; Eldan & Russinovich, 2023; Zhang et al., 2024; Dong et al., 2024; Fan et al., 2024). While prior work has found that these methods *can* be effective, they are generally costly and almost always result in utility loss.

**General inference-time approaches.** Soft-prompting and in-context learning (Muresanu et al., 2024; Pawelczyk et al., 2024; Bhaila et al., 2025) aim to also approximate the effects of unlearning by modifying the input to the model rather than the weights. However, these methods are still sensitive to changes in inputs e.g., they can be jailbroken easily, and the methods tend to be very niche/specialized use cases. There are many different approaches to placing classifiers or guardrails before and after the base model (Gao et al., 2025; Inan et al., 2023; Sharma et al., 2025), though these tend to be effectively binary measures to flag inappropriate inputs and outputs.

**Steering methods modify outputs during inference.** Activation-space steering computes a direction representing a conceptual contrast (e.g., "love" vs. "hate") and injects that vector during forward passes (Turner et al., 2024). This provides a way to push the model toward or away from certain behaviors but is static, resulting in weaknesses such as if applied to refusals it would have a very high false positive rate. (Lee et al., 2025) extend this line of work by making steering conditional: the steering vector is applied only when the input resembles a predefined concept, enabling targeted refusals without unnecessary over-refusal. Our work builds on this direction by allowing even more adaptive, model-aware steering that generalizes beyond safety and refusal behaviors. There is also a conceptual parallel to LLM watermarking (Dathathri et al., 2024; Li et al., 2025), which subtly biases generation trajectories while keeping outputs fluent. In contemporaneous work, Suriyakumar et al. (2025) empirically motivate inference-time unlearning via a linear setup based on the performance of a single TOFU metric and individual MUSE metrics. In contrast, we aggregate the MUSE metrics, recognizing the tradeoffs inherent in individual metric performance, and assess both linear

and rank-based divergence decoding across more than 20 different TOFU metrics following Dorna et al. (2025) and MUSE. We also perform extensive ablations on hyper-parameters such as model sizes, demonstrate the efficacy of n-gram based small models, profile the compute and runtime for over 1,200 model combinations, and explore the generalizability to domains beyond text.

**Smaller models do not necessarily imply a loss of performance.** Evidence from (Gunasekar et al., 2023; Bucher & Martini, 2024; Pecher et al., 2025) show that when finetuned for specialized tasks, small models can match or outperform the performance of general larger models. In addition, (Leviathan et al., 2023) proposed Speculative Decoding, demonstrating that smaller models can be used to accelerate inference in tandem with larger models. Contrastive Decoding (Li et al., 2023) also uses a smaller model in order to boost the performance of a larger model. Our work extends this literature by introducing a method of unlearning which relies on small specialized models to guide a larger model away from undesirable output.

## 3 METHOD

We begin by defining the problem, introducing our method, and finally connecting it to existing work. Let $V$ denote a finite vocabulary of tokens. A token sequence of length $T$ is denoted as $x = (x_1, x_2, ..., x_T)$ where each token $x_t \in V$. The prefix of a token sequence up to token $t-1$ is denoted $x_{<t} = (x_1, ..., x_{t-1})$. There are two data generating distributions $D_A$ and $D_B$ where the support of $D_B$ is contained within $D_A$. Finally, $P(x_t|x_{<t})$ and $Q(x_t|x_{<t})$ denote the conditional token distributions under $D_A$ and $D_B$, respectively.

We consider the situation where we wish to sample from $Q$ but do not have access to it. Instead, only $P$ is accessible. For example, $P$ could be a large frontier model for which it is cost prohibitive to retrain a new model from scratch on $D_B$. Within the finance domain, $Q$ could be a model as capable as $P$ but trained up to a fixed knowledge cutoff so as to avoid look-ahead bias. Generally, our goal is to approximate sampling from $Q$ using only $P$ and samples drawn from $D_A$ and $D_B$.

### 3.1 DIVERGENCE DECODING

Consider two small models $p(x_t|x_{<t})$ and $q(x_t|x_{<t})$ trained on samples from $D_A$ and $D_B$, respectively. Denote the logits of a given model $M$ as $l_M(x_{<t}) \in \mathbb{R}^{|V|}$. Divergence Decoding (DD) approximates sampling from $Q$ by adjusting the logits of $P$ according to the divergence between $q$ and $p$. Empirically, we consider two adjustments. The first is a linear combination of the logits,

$$\hat{l}_Q^{LC}(x_{<t}) = l_P(x_{<t}) + \alpha \cdot [l_q(x_{<t}) - l_p(x_{<t})], \tag{1}$$

while the second adjustment is rank based,

$$\hat{l}_Q^R(x_{<t}) = l_P(x_{<t}) - \mathbb{1}_{rank(l_p(x_{<t})-l_q(x_{<t})) \leq k} \cdot \infty. \tag{2}$$

In the case of the linear adjustment, if the difference between $Q$ and $P$ is indeed linear in logit space, then there exists some value of $\alpha$, $p$, and $q$ which enables $Q$ to be perfectly recovered. If the difference is not linear however, then this is not true. For this reason, we also explore the rank based approach, which prevents generating the top-$k$ most divergent tokens between $p$ and $q$.

Samples can then be drawn via typical methods (e.g., Fan et al., 2018; Holtzman et al., 2020) from the approximation,

$$\widehat{Q}(x_t|x_{<t}) = \text{softmax}(\hat{l}_Q(x_{<t})). \tag{3}$$

While the adjustments in Eq. 1 and 2 require additional forward passes for $p$ and $q$, we show in Section 4 that strong performance on certain tasks can be achieved even when $p$ and $q$ are trigram models—which add negligible computational overhead.

### 3.2 THEORETICAL MOTIVATION

While simple to implement and fast at inference-time, our method is theoretically motivated by the Product of Experts (Hinton, 1999) and Importance Sampling (Hammersley & Handscomb, 1965)

literature. In Appendix A, we show that the approximation $\widehat{Q}$ can be formulated as a Product of Experts model,

$$\widehat{Q}(x_t|x_{<t}) \propto \underbrace{P(x_t|x_{<t})}_{\text{Base Expert}} \cdot \underbrace{\left[\frac{q(x_t|x_{<t})}{p(x_t|x_{<t})}\right]^\alpha}_{\text{Domain Expert}} \quad (4)$$

where $\widehat{Q}$ is the product of a "Base Expert" $P$ responsible for providing foundational knowledge and a "Domain Expert" comprised of the ratio of $q$ to $p$. Intuitively, the role of the domain expert can be summarized by three cases:

1. $q \approx p$: Tokens are similarly likely under both $D_A$ and $D_B$ and the domain expert ratio is close to 1 effectively leaving the probabilities from the base model $P$ unchanged

2. $q \gg p$: Tokens are much **_more_** likely under $D_B$ than $D_A$, and the domain expert "upvotes" such tokens by **_increasing_** the probability assigned to them

3. $q \ll p$: Tokens are much **_less_** likely under $D_B$ than $D_A$, and the domain expert "downvotes" such tokens by **_decreasing_** the probability assigned to them

Finally, DD can also be linked to importance sampling in Monte Carlo analysis whereby the expectation of some function $f(x)$ under a target distribution $D_{target}$ is estimated using samples drawn from a proposal $D_{proposal}$. Formally,

$$\mathbb{E}_{x \sim D_{target}}[f(x)] = \mathbb{E}_{x \sim D_{proposal}}\left[f(x)\frac{D_{target}(x)}{D_{proposal}(x)}\right], \quad (5)$$

where the importance weight $w(x) = \frac{D_{target}(x)}{D_{proposal}(x)}$ adjusts the expectation taken over $D_{proposal}$ for differences between the proposal and target distributions. Analogously, divergence decoding uses the ratio of $q$ to $p$ to adjust for differences between the inaccessible model $Q$ and accessible one $P$.

## 4 BENCHMARKS

We evaluate our method on two standard unlearning benchmarks—MUSE and TOFU—using the Open Unlearning framework (Dorna et al., 2025; Maini et al., 2024; Shi et al., 2024). Following the MUSE vocabulary, the **Target** model refers to the model subject to unlearning, while **Retrain** denotes the best—but most costly—baseline obtained by retraining from scratch.

We fine-tune one model on the retain set and one on the forget set. To avoid excessive divergence between $p$ and $q$, the forget model may also include retain data when the retain set is substantially larger. In the MUSE news dataset, the retain set is roughly twice the size of the forget set, and training the forget model only on the forget data performs best. In contrast, for TOFU's '90' benchmark—where the retain set is nine times larger—training on both forget and retain works best.

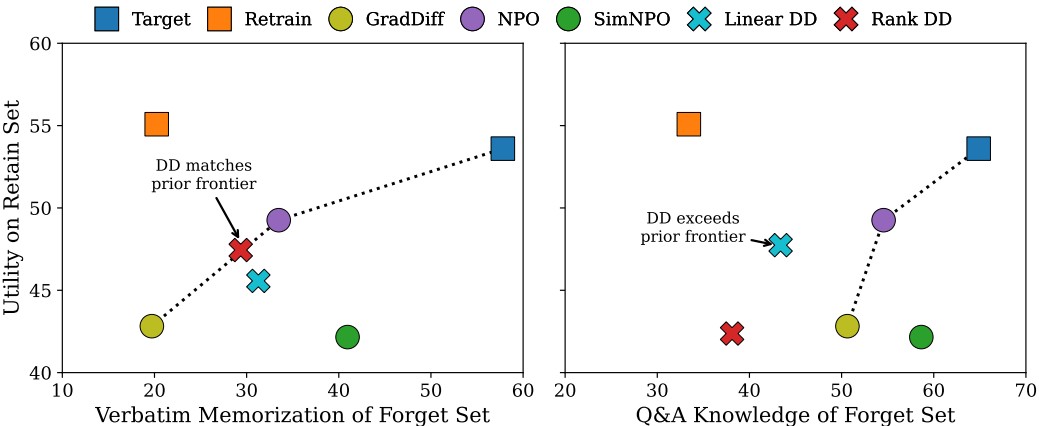

Figure 1: MUSE Results. Closer to retrain is better. 99% CIs are smaller than the marker sizes.

Table 1: TOFU Results

| Method | Config | Agg. ↑ | Mem. ↑ | Priv. ↑ | Utility ↑ |
|---|---|---|---|---|---|
| Target | Full | $0.02 \pm 0.01$ | $0.01 \pm 0.00$ | $0.38 \pm 0.00$ | $1.00 \pm 0.04$ |
| Retrain | Retain90 | $0.78 \pm 0.01$ | $0.53 \pm 0.02$ | $0.98 \pm 0.01$ | $1.03 \pm 0.04$ |
| Linear DD | $\alpha$=1.5 | $0.78 \pm 0.01$ | $0.56 \pm 0.02$ | $0.95 \pm 0.03$ | $1.00 \pm 0.04$ |
| Rank DD | k=20 | $0.85 \pm 0.02$ | $0.80 \pm 0.01$ | $0.81 \pm 0.02$ | $0.95 \pm 0.05$ |
| DPO | lr=4e-6, e=2 | $0.31 \pm 0.10$ | $0.21 \pm 0.01$ | $0.39 \pm 0.00$ | $0.43 \pm 0.29$ |
| GradAscent | lr=2e-6, e=3 | $0.63 \pm 0.01$ | $0.51 \pm 0.01$ | $0.61 \pm 0.02$ | $0.87 \pm 0.04$ |
| GradDiff | lr=2e-6, e=3 | $0.64 \pm 0.01$ | $0.52 \pm 0.01$ | $0.62 \pm 0.02$ | $0.86 \pm 0.04$ |
| NPO | lr=4e-6, e=2 | $0.67 \pm 0.01$ | $0.57 \pm 0.01$ | $0.68 \pm 0.02$ | $0.82 \pm 0.03$ |
| RMU | lr=8e-7, e=4 | $0.67 \pm 0.02$ | $0.60 \pm 0.01$ | $0.74 \pm 0.03$ | $0.69 \pm 0.04$ |

Note: Agg. is the harmonic mean of Mem., Priv., and Utility. Each of these is itself the harmonic mean of several tests. The top entry per column is boldfaced. See Appendix F of Dorna et al. (2025) for details on the construction of these metrics. 99% CIs computed via hierarchical bootstrap resampling

For the MUSE benchmark (Shi et al., 2023; 2025), we use the news dataset and finetune *princeton-nlp/Sheared-LLaMA-1.3B* (Xia et al., 2023) for both $p$ and $q$. This model shares its tokenizer and training data distribution with the official MUSE models. As seen in Figure 1, our method matches or exceeds unlearning methods across memorization and Q&A dimensions.

For TOFU (Maini et al., 2024), we use the LLaMA 3.2 1B *retain90* and *full* models as $p$ and $q$, respectively, and the LLaMA 3.1 8B model as $P$. As summarized in Table 1, using $\alpha$ slightly larger than 1 yields an almost perfect approximation of the behavior of the retrain model.

## 5 ABLATIONS

For our study of hyper-parameter choice, algorithm choice, and model size for MUSE, we consider euclidean distance to Retrain, normalized such that Target is 100%, as our all encompassing score to capture the utility and forgetting tradeoff, for both Q&A and memorization. For TOFU, we simply consider the aggregate score with and without privacy, as discussed in Appendix F of Dorna et al. (2025). It is important to note that TOFU uses instruction tuned models while MUSE uses only pre-trained models with few-shot Q&A and significantly simpler questions and answers. In addition, the datasets used in MUSE are much larger. We keep the model training setup fixed; in principle, further fine-tuning would allow smaller hyperparameter values. Figure 9 contains the raw data points used in the ablation studies for MUSE, and Table 7 contains the raw information for TOFU.

### 5.1 HYPER-PARAMETER CHOICE AND ALGORITHM CHOICE

We first study the choice of Linear DD vs Rank DD and the sensitivity to hyper-parameter choice. On MUSE, we find that Rank DD outperforms on memorization while Linear DD marginally outperforms on Q&A. On TOFU, Rank DD marginally outperforms Linear DD on the aggregate metric, both when privacy is included and when it is excluded. We find that there are a large range of hyper-parameter values that perform well and that Rank DD is especially flexible.

### 5.2 MODEL SIZE

Given that our method works well with the 1B and 1.3B small models, a natural question is how sensitive performance is to the size of $p$ and $q$. We investigate this using the the 2.7B Sheared-LLaMA model variant for MUSE, the Llama 3.2 3B variants for TOFU models, and trigram LMs based on *Stupid Backoff* (Brants et al., 2007) for both. Our trigram implementation pre-computes all scores as arrays the size of the vocabulary, effectively giving zero inference overhead and serving as a limiting case of "0% of $P$'s size."

We evaluate the most optimal configuration for each model size. On MUSE, scaling from 1.3B to 2.7B yields a noticeably larger gain than the corresponding jump from 1B to 3B on TOFU. Meanwhile, the trigram models—which perform surprisingly well on some MUSE settings—fail almost

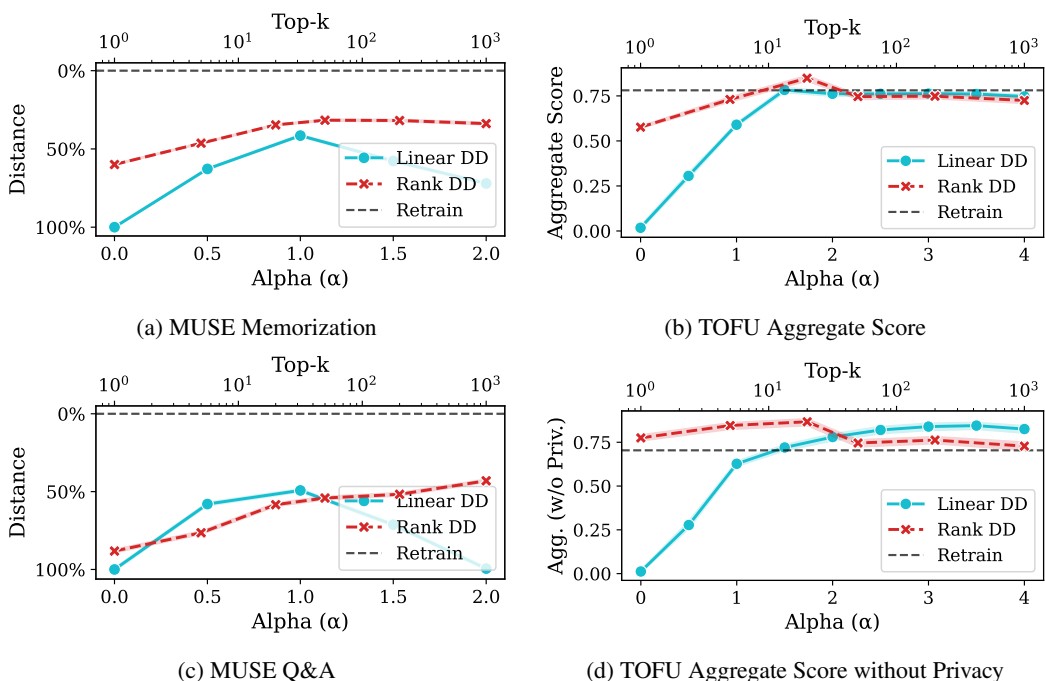

Figure 2: Effect of hyper-parameter and algorithm choice. The Alpha scale runs from 0 to 2 on MUSE and 0 to 4 on TOFU. 99% CI are provided.

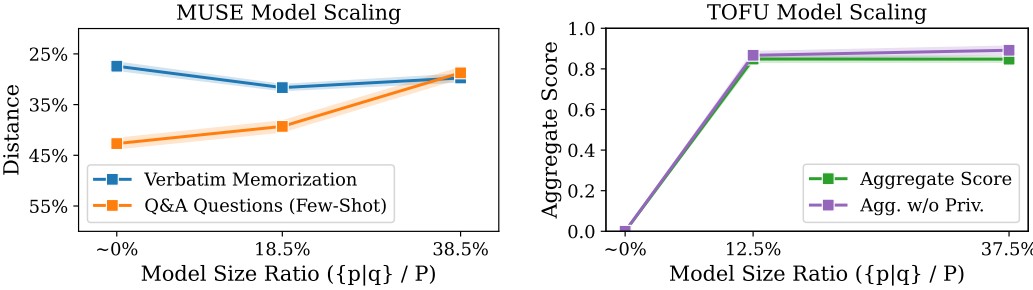

Figure 3: Analysis of model scaling on MUSE (left) and TOFU (right). 99% CI are provided.

entirely on TOFU. Upon further inspection of the Q&A questions on MUSE where the Trigram models perform well, we find that this is largely due to questions which are more similar to the underlying training data. Thus, we conclude that the Trigram models are likely most useful for unlearning verbatim content.

### 5.3 Over-Unlearning, Privacy, and Calibration

**Over-unlearning, even when utility is preserved, is not always optimal.** In settings like toxic content prevention, aggressively suppressing certain outputs is entirely reasonable. However, many real-world applications are highly sensitive to *over*-unlearning. For instance, in financial modeling—such as backtesting trading strategies or stress testing banks—the goal is to evaluate performance using only the information that would have been available at the time. For example, one would want to unlearn the 2008 financial crisis so they could realistically assess the performance of an LLM making decisions at the time. **Over-unlearning would cause the model to overcompensate** to the point that it assigns even lower likelihoods to these events than what ultimately occurred.

More broadly, we treat the privacy metrics as indicators of over- versus under-unlearning, rather than as definitive tests of whether individual training examples were used or successfully removed. As

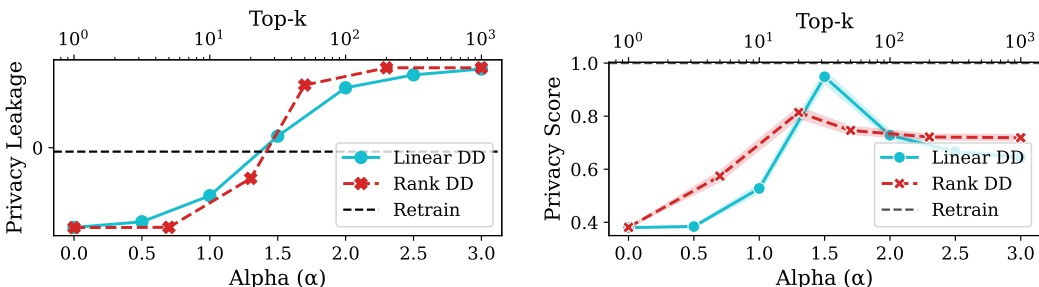

Figure 4: Analysis of Over- or Under- Unlearning on MUSE (left) and TOFU (right). Closer to retrain is better. The optimal values for both benchmarks are Alpha~1.5 and TopK~20. 99% CI are provided.

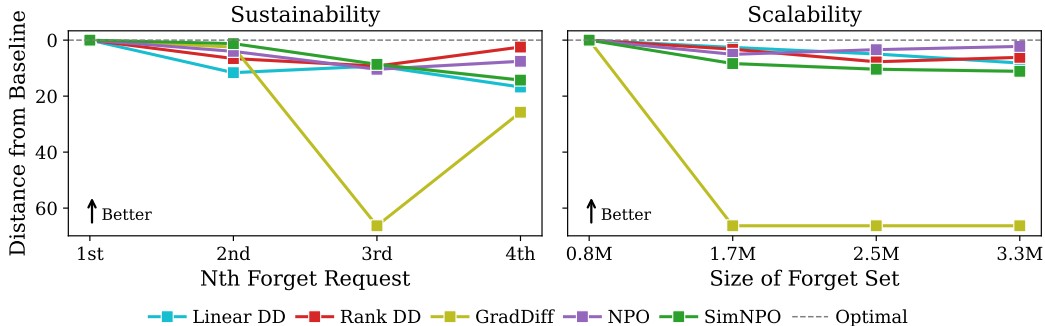

Figure 5: The left column is sustainability - consecutive forget sets of the same size - and the right column is scaling, increasingly large forget sets. We consider euclidean distance to the method's baseline performance when evaluated on the retain set and the **original** forget set, with the increasing distance capturing both **utility loss** and **loss of forgetting.** In general, all methods except for GradDiff perform reasonably well and within the margin of error of each other.

discussed in Section 8, these methods are not intended for open-source distributions of the weights, though it can be used if the logits are public. The naive implementation of the *rank based method*—e.g., setting targeted logits to $-\infty$—would produce degenerate privacy scores, since the losses would be infinite. To preserve the ability to evaluate over- versus under-unlearning in the rank-based setting, we instead replace the $k$ most divergent logits with the $k$th largest logit in the unmodified distribution.

Across both MUSE and TOFU, a broad range of hyper-parameters produce models that are statistically **indistinguishable** from a full retrain, striking a clean balance between over- and under-unlearning. The fact that the optimal region occurs around $\alpha > 1$ aligns with the intuition from §3.1 that a simple linear combinations of logits may be a near-optimal solution.

## 5.4 SUSTAINABILITY AND SCALING

Finally, prior work has found that many unlearning methods exhibit poor scalability—the unlearning of very large amounts of content—and sustainability—sequential requests to unlearn additional content. We explore the efficacy of our method along these dimensions using the MUSE scaling and sustainability benchmarks to ensure that performance does not degrade. To extend the benchmark, we additionally measure performance on the original forget set (Q&A), ensuring that larger and subsequent forget requests **do not** come at the cost of the forget weights being overwritten.

## 6 BEYOND TEXT

One benefit of our method is its generality, i.e., it can be applied to any setting where samples are drawn from some distribution $P$ and data exists to estimate $p$ and $q$. Along these lines, we explore the extent to which our method is effective in domains beyond text by applying it to image generation.

We begin with the setup of Esser et al. (2021) and augment the sampling in latent space per equations 1 and 2. The models $p$ and $q$ are estimated using data from the train split of ImageNet associated with the dog synset. Specifically, half the descendants from the dog synset are randomly assigned to the forget set $F$ and the other half to the retain set $R$—notably, this random assignment ensures that any preferences over dog classes will be uncorrelated with the assignment to retain versus forget. The class-conditional ImageNet checkpoint from Esser et al. (2021) is then fine-tuned on $F$ and $R$ to estimate $p$ and $q$, respectively. We then sample images from the model configured without any divergence decoding (Baseline) and with various linear and rank-based (Figure 6 and Appendix D). As a first quantitative evaluation of the efficacy of our method, we evaluate the content of class

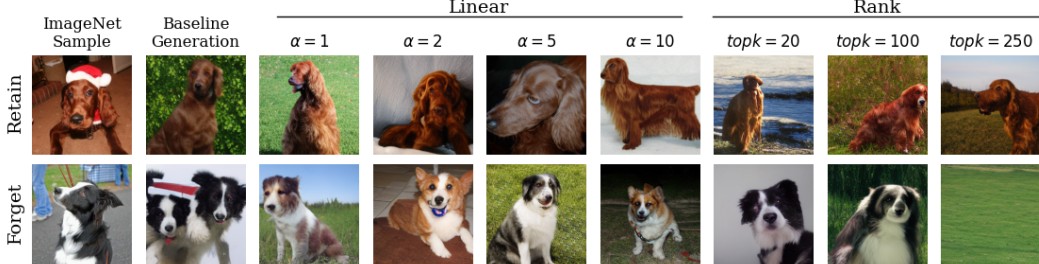

Figure 6: ImageNet examples, baseline generations, and generations under various divergence decoding setups.

conditional generations using VQAScore (Lin et al., 2024). For each class conditional generated image, we prompt a multi-modal LLM (MLLM) to assess whether the image contains the specific class and take the probability of "Yes" as the VQAScore—we rely on GPT-4o-mini for this task as it requires access to the log probabilities and many modern closed models do not provide this, e.g., GPT-5-nano. Table 2 presents mean VQAScores for class conditional samples split by whether the class was assigned to the retain or forget set. For linear divergence decoding setups, modest settings of alpha display efficacy, e.g., $\alpha = 1$ decreases the mean VQAScore on classes in the forget set from 97% to 20%. This is similar to our findings within the text domain where $\alpha$ in the range of 1 to 2 typically yielded the best results. In contrast, the rank-based setups require larger values for top-k to reach similar efficacy, e.g., $topk = 250$.

As a second evaluation of the efficacy of our method, we evaluate the perceptual quality of generated images. Notably, a naive unlearning method could simply output noise for classes in the forget set. While this would constitute "unlearning," it may not be particularly useful if the desired outcome is perceptually similar and plausible generations without the *indicia* associated with the classes to be forgotten, e.g., the identifiable style attributable to an artist requesting that a model provider adhere to copyright laws. Along these lines, we follow Chen et al. (2024) and employ an MLLM-as-a-judge to perform pairwise comparison of the visual quality between samples from our baseline setup and a given divergence decoding setup.

Table 3 presents the performance for a variety of MLLM judges and divergence decoding setups. In general, there is little decrease in the perceptual quality on samples conditional on classes in the retain set. For those in the forget set, however, there is a decrease in quality. For example, a setup with $\alpha = 5$ decreases the rate at which a generated image contains the class to be forgotten from 97% to 1%, but these images are also only preferred over baseline generations 31% of the time. As such, in practice one would have to sample, on average, two generations to get a sample which both does not contain the class to be forgotten and meets or exceeds the baseline quality.

Table 2: Content analysis of images generated using various divergence decoding setups. Mean values and standard errors are presented. GPT-4o-mini is used as a judge.

| Method | Config | Retain | Forget |
|---|---|---|---|
| Baseline | — | $96\% \pm 1.1$ | $97\% \pm 1.0$ |
| Linear | $\alpha = 1$ | $96\% \pm 1.2$ | $20\% \pm 2.4$ |
| Linear | $\alpha = 2$ | $97\% \pm 1.0$ | $20\% \pm 2.3$ |
| Linear | $\alpha = 5$ | $96\% \pm 1.1$ | $1\% \pm 0.6$ |
| Linear | $\alpha = 10$ | $96\% \pm 1.2$ | $1\% \pm 0.6$ |
| Rank | $topk = 20$ | $96\% \pm 1.1$ | $77\% \pm 2.5$ |
| Rank | $topk = 100$ | $95\% \pm 1.3$ | $58\% \pm 2.9$ |
| Rank | $topk = 250$ | $95\% \pm 1.2$ | $20\% \pm 2.4$ |

Table 3: Perceptual quality analysis of images generated using various divergence decoding setups and MLLM judges. Mean values and standard errors are presented.

| Method | Config | Gemini 2.5 Flash-Lite | | GPT-5-nano | | Qwen3-VL 8B | |
|---|---|---|---|---|---|---|---|
| | | Retain | Forget | Retain | Forget | Retain | Forget |
| Linear | $\alpha = 1$ | $52\% \pm 1.7$ | $38\% \pm 1.4$ | $50\% \pm 1.6$ | $38\% \pm 1.4$ | $50\% \pm 1.6$ | $36\% \pm 1.3$ |
| Linear | $\alpha = 2$ | $49\% \pm 1.6$ | $37\% \pm 1.3$ | $49\% \pm 1.6$ | $38\% \pm 1.4$ | $49\% \pm 1.6$ | $39\% \pm 1.4$ |
| Linear | $\alpha = 5$ | $52\% \pm 1.6$ | $31\% \pm 1.2$ | $49\% \pm 1.6$ | $31\% \pm 1.3$ | $52\% \pm 1.6$ | $31\% \pm 1.3$ |
| Linear | $\alpha = 10$ | $47\% \pm 1.6$ | $31\% \pm 1.3$ | $47\% \pm 1.6$ | $32\% \pm 1.3$ | $50\% \pm 1.6$ | $32\% \pm 1.2$ |
| Rank | $topk = 20$ | $49\% \pm 1.6$ | $47\% \pm 1.6$ | $48\% \pm 1.6$ | $47\% \pm 1.6$ | $48\% \pm 1.6$ | $48\% \pm 1.6$ |
| Rank | $topk = 100$ | $47\% \pm 1.6$ | $50\% \pm 1.5$ | $45\% \pm 1.6$ | $46\% \pm 1.5$ | $45\% \pm 1.6$ | $45\% \pm 1.5$ |
| Rank | $topk = 250$ | $48\% \pm 1.6$ | $20\% \pm 1.1$ | $49\% \pm 1.6$ | $21\% \pm 1.1$ | $46\% \pm 1.6$ | $21\% \pm 1.1$ |

## 7 COST AND LATENCY ANALYSIS

A key consideration of applying our method is the increased inference-time compute from running the two small models in tandem with the large model. Denote the number of parameters in the large model as $N$ and $n$ the number in each small model. Following the approximation of Kaplan et al. (2020), the total inference cost increases from $2N \longrightarrow 2(N + 2n)$, with the relative increase given by $2n/N$. In Figure 7, we empirically measure the increase in compute costs associated with running over 1,200 different combinations of models in a distributed setting within a single $8x\{H100|B200\}$ instance (see Appendix B for details). We find that the compute costs tend to scale closely with our theoretical approximation. In Appendix B, we examine the effect of our method on latency and find that the increase is generally less than 0.1%.

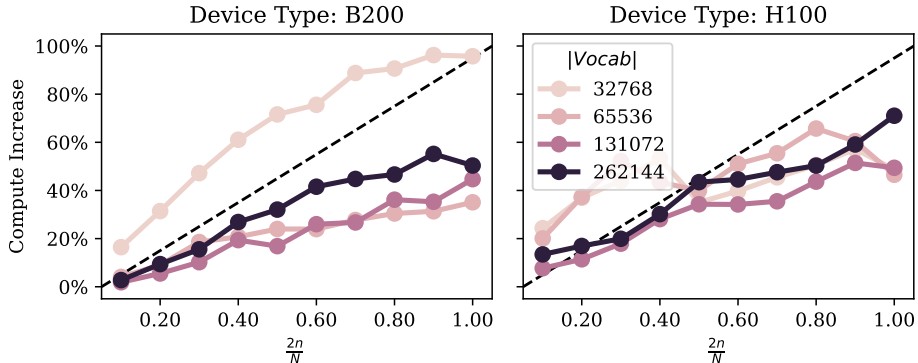

Figure 7: Empirical increases in compute requirements for a sample of more than 1,200 models. Size of $P$ ranges from 300M to 80B.

## 8    LIMITATIONS

A key limitation lies in the method's **sensitivity to instruction-tuning**. For instance, when unlearning financial knowledge, the model may generate stock recommendations in the format:

"**1. {firm name}:**"

If the smaller models anticipate a different structure (e.g., a ticker symbol or bullet marker after the '1.'), the divergence in logits at the critical step may be diluted or entirely noisy. Worse, if one small model aligns closely with the large model while the other does not, differences fail to cancel and can yield unstable or unintended outputs. Independent researchers adopting this method may therefore need to carefully re-tune instruction following behavior using publicly available datasets after modifying training mixtures, while in house researchers may not find this to be a problem. In general we expect this to be used by model providers for API-locked models.

Finally, DD does not erase internal representations; it only constrains outputs at decode time. This makes it unsuitable for preventing toxic or copyrighted generations in **open-weight settings**, since releasing the forget model's weights could reveal sensitive information. However, for API locked models it is still possible to expose the final logits or log-probabilities. For Linear DD, the resulting logits should be virtually indistinguishable from the base model. For rank DD, there may be strategies beyond the naive implementations of the method—such as masking with random samples, or only adjusting logits that are in the top-p/top-k of the original distribution—to safely make the logits indistinguishable.

## 9    CONCLUSION

In this work, we introduce a method to simulate unlearning at inference-time for selectively removing information from Large Language Models without costly retraining or fine-tuning of the base model. Our method, Divergence Decoding, leverages smaller, specialized models to guide text generation away from undesirable content at the point of inference. Our experiments demonstrate three key contributions. First, our approach is highly effective, significantly reducing the model's ability to recall both verbatim and semantic knowledge from a designated "forget set." Second, by confining training to small secondary models, our method offers a dramatically more efficient and scalable solution than machine unlearning, reducing computational overhead by orders of magnitude compared to existing techniques. Finally, because the weights of the large base model remain untouched, our method excels at utility preservation, maintaining performance on general knowledge benchmarks even as the number of unlearning requests grows. By providing a practical, low-cost, and effective solution to a critical challenge in AI safety and privacy, divergence decoding can potentially enable more responsible and adaptable deployment of large-scale language models.

## REPRODUCIBILITY STATEMENT

We took care to modify OpenUnlearning as little as possible, and have details about our setups for MUSE and TOFU in Appendix C. All code to reproduce this work is available at the following anonymous repository link: https://anonymous.4open.science/r/inference-time-unlearning-iclr2026/

We will release fine-tuned models and additional data on Hugging Face after the review period.

## ETHICS STATEMENT

In general, we intend unlearning to support beneficial use cases - for debiasing models, preventing toxic and copyrighted content generation, and legitimate research in domains such as finance. However, we acknowledge the approach could be misused to induce undesirable or harmful biases.

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

## A CONNECTION TO PRODUCT OF EXPERTS

Hinton (1999) introduced the Product of Experts (PoE) framework whereby $n$ probability models are multiplicatively combined into a single model. Let the $i$-th expert be denoted by $f_i(x|\theta_i)$, then a PoE model $R$ comprised of $n$ experts is given by,

$$R(x|\theta_1, ..., \theta_n) = \frac{1}{Z} \prod_{i=1}^{n} f_i(x|\theta_i), \tag{6}$$

where $Z$ is a normalization constant. To highlight the connection between divergence decoding and PoE, recall Eq. 1:

$$\hat{l}_Q(x_{<t}) = l_P(x_{<t}) + \alpha \cdot [l_q(x_{<t}) - l_p(x_{<t})].$$

In Eq. 1, a given model $M$ has logits which are equal to the log-probabilities up to an additive constant which depends on the token sequence prefix $x_{<t}$ but not the token $x_t$, i.e.,

$$l_M(x_{<t}) = \log M(x_t|x_{<t}) + C_M(x_{<t}). \tag{7}$$

Substituting Eq. 7 into Eq. 1 for each model, gathering the constants, and performing some algebra reveals the link to PoE:

$$\log \widehat{Q}(x_t|x_{<t}) = \log P(x_t|x_{<t}) + \alpha \cdot [\log q(x_t|x_{<t}) - \log p(x_t|x_{<t})] + C$$

$$\widehat{Q}(x_t|x_{<t}) \propto \exp \left( \log P(x_t|x_{<t}) + \alpha \cdot [\log q(x_t|x_{<t}) - \log p(x_t|x_{<t})] \right)$$

$$\propto P(x_t|x_{<t}) \cdot q(x_t|x_{<t})^{\alpha} \cdot p(x_t|x_{<t})^{-\alpha}$$

$$\propto P(x_t|x_{<t}) \cdot \left[ \frac{q(x_t|x_{<t})}{p(x_t|x_{<t})} \right]^{\alpha}.$$

## B DETAILED ANALYSIS OF COMPUTE AND RUNTIME COSTS

In this section we explore the compute and runtime costs associated with our method using theoretical and empirical analyses. Additionally, we compare these costs to those associated with other unlearning methods to provide guidance on when our method is desirable. In general, we find that our method introduces minimal latency (less than 0.1% increases in realistic production environments) and compares favorably to other unlearning methods for a wide range of compute budgets.

### B.1 COMPUTE REQUIREMENTS

As presented in Section 7, the compute increase associated with our method can be approximated as $2n/N$ where $n$ and $N$ are the number of parameters in the small and big models, respectively. For example, applying our method to a 10B parameter model using 1B parameter small models is expected to require a 20% increase in compute. While this is a useful theoretical approximation, we empirically explore this approximation using a distributed setup on 8xH100 and 8xB200 instances using more than 1,200 unique combinations of models for $P$, $p$, and $q$.

For our specific setup, we target an environment where the small models $p$ and $q$ are running on some number of accelerators while multiple copies of the large model $P$ is running on additional accelerators. We consider the set of candidate models for $P$, $p$, and $q$ as those listed in Table A9 of Hoffmann et al. (2022) and add several models in the range of 19-70B parameters following the Llama 3 architecture (Grattafiori et al., 2024). Additionally, we consider four vocabulary sizes for each model: $2^{15}$, $2^{16}$, $2^{17}$, and $2^{18}$.

We then match models for $p$ and $q$ to $P$ such that $2n \leq N$ and only consider models for $P$ where $N > 8e9$. The compute increase required to run a given combination of models is then measured as the ratio $(t_P + t_p + t_q)/t_P$ where $t$ is the time required to run the models measured in GPU-hrs. Results for all combinations of models, vocabulary sizes, and devices are presented in Figure 7. In general, we find a strong agreement with the theoretical approximation.

## B.2 LATENCY

An additional consideration of applying our method is latency, i.e., many applications require fast responses to users and the an increase in latency of 10-20% could be unacceptable. Following from Section B.1 above, we explore the latency impact of our method in a distributed environment where the small models $p$ and $q$ can be run in parallel with multiple copies of $P$. In this setting, the primary contributor to latency is the time required to sync the logits in Eq. 1 across devices such that sampling from the approximation to $Q$ can be performed.

Along these lines, we measure the increase in runtime as the ratio $t_Q/t_P$ where the time $t$ is the total time required to generate a sequence of fixed length, $t_Q$ is the time to do this under the distributed divergence decoding setup, and $t_P$ is the time to do this under a setup where $P$ is run on a single GPU with no synchronization overhead. The two key factors here are the vocabulary size which determines the size of the data being synchronized across GPUs and the size $N$ of $P$ which determines the baseline runtime required. Figure 8 shows that for most model configurations, the increase in runtime is less than 0.1%. For smaller "large" models, i.e., $N < 20e9$, and the largest vocabulary size, the increase is in runtime is roughly 0.2-0.5%. Thus, while there is undeniably an increase in latency, it is relatively modest at $< 0.5\%$ for the vast majority of realistic model configurations.

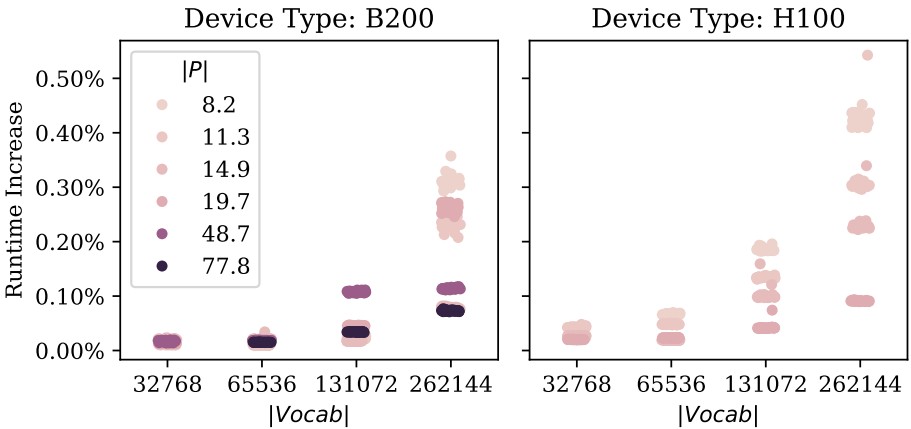

Figure 8: Effect of model and vocabulary size on runtime for two generations of accelerators

## B.3 RELATIVE TO OTHER UNLEARNING METHODS

Additionally, let $d_r$ and $d_f$ be the sizes of the retain and forget datasets (in tokens), let $e_N$ and $e_n$ be the number of epochs the large and small models are trained for, respectively, and let $I$ be the number of inference tokens. Hence, we want to know after how many inference tokens does it become more costly to use DD over another method, **assuming both work equally well**. Considering one of the simplest unlearning methods, Gradient Ascent (Jang et al., 2023) **without any kind of regularizer**, DD becomes more costly once:

$$6ne_n(d_r + d_f) + 2(N + 2n)I \geq 6Ne_N(d_f) + 2NI$$
$$I \geq \frac{3Ne_N d_f}{2n} - \frac{3e_n(d_r + d_f)}{2}$$

## C DETAILED EXPERIMENTAL SETUPS

### C.1 MUSE

We finetune the LlaMA models using the **AdamW Torch optimizer** and a **cosine scheduler** for **10** epochs. We set the learning rate such that the loss approximately halves over the

course of training. We swept the LLaMA models with $\alpha \in \{0.5, 0.6, \ldots, 1.5\}$ and top-$k \in \{1, 5, 20, 50, 100, 200, 500, 1000\}$ and the trigram models at $\alpha \in \{5, 10, \ldots 30\}$ and top-$k \in \{1, 2, 3, 5, 10\}$.

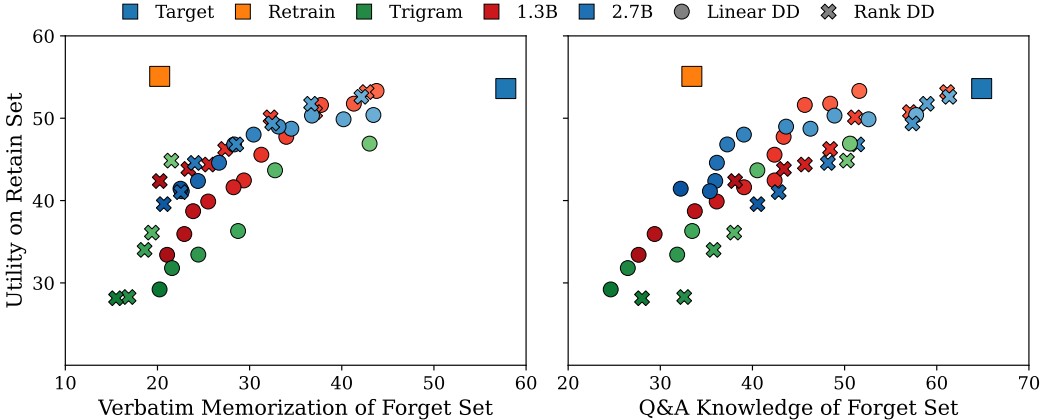

Figure 9: All hyper-parameter and model size configurations. Increasing values are darker and usually with reduced scores on both utility and memorization (to the bottom and left.)

Table 4: Configuration MUSE

| Model | Initial LR | Best Verbatim | Best Q&A |
|---|---|---|---|
| Stupid Backoff Trigram | | TopK=1 | Alpha=10 |
| princeton-nlp/Sheared-LLaMA-1.3B | 5e-5 | TopK=100 | Alpha=0.8 |
| princeton-nlp/Sheared-LLaMA-2.7B | 4e-5 | TopK=200 | Alpha=1.0 |

For the other methods, we use the default settings provided by OpenUnlearning

Table 5: MUSE Configurations

| Method | Epochs | Method-Specific Hyperparameters |
|---|---|---|
| GradDiff | 1* | $\alpha = 1.0, \ \gamma = 1.0$ |
| NPO | 10 | $\beta = 0.1, \ \alpha = 1.0, \ \gamma = 1.0$ |
| SimNPO | 10 | $\delta = 0, \ \beta = 4.5, \ \alpha = 1.0, \ \gamma = 0.125$ |

Default hyperparameters: batch size = 32, learning rate = $1 \times 10^{-5}$, warmup epochs = 1, weight decay = 0.01, retain loss = NLL. * For GradDiff, the 1 epoch setting is the only deviation from the defaults.

## C.2 TOFU

For $p$ and $q$ we use *open-unlearning/tofu_Llama-3.2-1B-Instruct_full*, *open-unlearning/tofu_Llama-3.2-1B-Instruct_retain90*, and the counterparts for 3B. For the other methods, we grid search learning rates $\{5 \times 10^{-7}, 8 \times 10^{-7}, 1 \times 10^{-6}, 2 \times 10^{-6}, 3 \times 10^{-6}, 4 \times 10^{-6}, 5 \times 10^{-6}, 1 \times 10^{-5}\}$ and epochs from 1 to 10. Below, we summarize the default hyperparameters provided by OpenUnlearning.

Table 6: TOFU Default Configurations (defaults apply unless noted)

| Method | Method-Specific Hyperparameters |
|---|---|
| DPO | $\beta = 0.1,\ \alpha = 1.0,\ \gamma = 1.0,$ retain loss = NLL |
| GradAscent | N/A |
| GradDiff | $\alpha = 1.0,\ \gamma = 1.0,$ retain loss = NLL |
| NPO | $\beta = 0.1,\ \alpha = 1.0,\ \gamma = 1.0,$ retain loss = NLL |
| RMU | $\alpha = 1.0,\ \gamma = 1.0,$ steering coef = 2, retain loss = Embed Diff |

Default (shared) hyperparameters: batch size = 32, warmup epochs = 1, weight decay = 0.01.

Table 7: All TOFU Divergence Decoding Results

| Size | Method | Param | Agg. ↑ | Agg w/o Priv. ↑ | Mem. ↑ | Priv. ↑ | Utility ↑ |
|------|--------|-------|--------|-----------------|--------|---------|-----------|
| 1B | Linear | α=0.5 | $0.31 \pm 0.02$ | $0.28 \pm 0.02$ | $0.16 \pm 0.01$ | $0.38 \pm 0.00$ | $1.00 \pm 0.04$ |
| 1B | Linear | α=1.0 | $0.59 \pm 0.01$ | $0.63 \pm 0.02$ | $0.46 \pm 0.01$ | $0.53 \pm 0.02$ | $1.00 \pm 0.04$ |
| 1B | Linear | α=1.1 | $0.64 \pm 0.01$ | $0.65 \pm 0.02$ | $0.48 \pm 0.01$ | $0.63 \pm 0.02$ | $1.00 \pm 0.04$ |
| 1B | Linear | α=1.2 | $0.69 \pm 0.01$ | $0.67 \pm 0.02$ | $0.51 \pm 0.01$ | $0.74 \pm 0.03$ | $1.00 \pm 0.04$ |
| 1B | Linear | α=1.3 | $0.74 \pm 0.01$ | $0.69 \pm 0.02$ | $0.53 \pm 0.01$ | $0.86 \pm 0.03$ | $1.00 \pm 0.04$ |
| 1B | Linear | α=1.4 | $0.77 \pm 0.02$ | $0.71 \pm 0.02$ | $0.55 \pm 0.02$ | $0.96 \pm 0.02$ | $1.00 \pm 0.04$ |
| 1B | Linear | α=1.5 | $0.78 \pm 0.01$ | $0.72 \pm 0.02$ | $0.56 \pm 0.02$ | $0.95 \pm 0.03$ | $1.00 \pm 0.04$ |
| 1B | Linear | α=2.0 | $0.76 \pm 0.01$ | $0.78 \pm 0.02$ | $0.65 \pm 0.02$ | $0.73 \pm 0.02$ | $0.99 \pm 0.04$ |
| 1B | Linear | α=2.5 | $0.76 \pm 0.01$ | $0.82 \pm 0.02$ | $0.71 \pm 0.02$ | $0.67 \pm 0.01$ | $0.97 \pm 0.05$ |
| 1B | Linear | α=3.0 | $0.76 \pm 0.01$ | $0.84 \pm 0.02$ | $0.76 \pm 0.02$ | $0.64 \pm 0.01$ | $0.93 \pm 0.05$ |
| 1B | Linear | α=3.1 | $0.76 \pm 0.01$ | $0.84 \pm 0.02$ | $0.77 \pm 0.02$ | $0.64 \pm 0.01$ | $0.92 \pm 0.04$ |
| 1B | Linear | α=3.2 | $0.76 \pm 0.01$ | $0.84 \pm 0.02$ | $0.78 \pm 0.02$ | $0.64 \pm 0.01$ | $0.92 \pm 0.04$ |
| 1B | Linear | α=3.3 | $0.76 \pm 0.01$ | $0.85 \pm 0.02$ | $0.79 \pm 0.02$ | $0.64 \pm 0.01$ | $0.91 \pm 0.04$ |
| 1B | Linear | α=3.4 | $0.76 \pm 0.01$ | $0.84 \pm 0.02$ | $0.80 \pm 0.02$ | $0.64 \pm 0.01$ | $0.90 \pm 0.04$ |
| 1B | Linear | α=3.5 | $0.76 \pm 0.01$ | $0.85 \pm 0.02$ | $0.80 \pm 0.02$ | $0.63 \pm 0.01$ | $0.89 \pm 0.04$ |
| 1B | Linear | α=4.0 | $0.75 \pm 0.01$ | $0.83 \pm 0.02$ | $0.83 \pm 0.01$ | $0.63 \pm 0.01$ | $0.82 \pm 0.04$ |
| 3B | Linear | α=0.5 | $0.39 \pm 0.01$ | $0.38 \pm 0.02$ | $0.24 \pm 0.01$ | $0.39 \pm 0.00$ | $1.01 \pm 0.04$ |
| 3B | Linear | α=1.0 | $0.70 \pm 0.01$ | $0.68 \pm 0.02$ | $0.52 \pm 0.01$ | $0.73 \pm 0.02$ | $1.00 \pm 0.04$ |
| 3B | Linear | α=1.1 | $0.76 \pm 0.01$ | $0.70 \pm 0.02$ | $0.54 \pm 0.02$ | $0.89 \pm 0.03$ | $1.00 \pm 0.04$ |
| 3B | Linear | α=1.2 | $0.79 \pm 0.02$ | $0.72 \pm 0.02$ | $0.56 \pm 0.02$ | $0.97 \pm 0.02$ | $1.00 \pm 0.04$ |
| 3B | Linear | α=1.3 | $0.78 \pm 0.02$ | $0.73 \pm 0.02$ | $0.58 \pm 0.02$ | $0.88 \pm 0.02$ | $1.00 \pm 0.05$ |
| 3B | Linear | α=1.4 | $0.76 \pm 0.01$ | $0.75 \pm 0.02$ | $0.60 \pm 0.02$ | $0.80 \pm 0.02$ | $1.00 \pm 0.05$ |
| 3B | Linear | α=1.5 | $0.76 \pm 0.01$ | $0.76 \pm 0.02$ | $0.62 \pm 0.02$ | $0.75 \pm 0.02$ | $0.99 \pm 0.04$ |
| 3B | Linear | α=2.0 | $0.76 \pm 0.01$ | $0.82 \pm 0.02$ | $0.70 \pm 0.02$ | $0.66 \pm 0.01$ | $0.97 \pm 0.04$ |
| 3B | Linear | α=2.5 | $0.76 \pm 0.01$ | $0.84 \pm 0.02$ | $0.76 \pm 0.02$ | $0.64 \pm 0.01$ | $0.94 \pm 0.04$ |
| 3B | Linear | α=2.6 | $0.76 \pm 0.01$ | $0.85 \pm 0.02$ | $0.77 \pm 0.02$ | $0.64 \pm 0.01$ | $0.94 \pm 0.04$ |
| 3B | Linear | α=2.7 | $0.77 \pm 0.01$ | $0.85 \pm 0.02$ | $0.78 \pm 0.02$ | $0.64 \pm 0.01$ | $0.94 \pm 0.04$ |
| 3B | Linear | α=2.8 | $0.77 \pm 0.01$ | $0.86 \pm 0.02$ | $0.79 \pm 0.02$ | $0.63 \pm 0.01$ | $0.93 \pm 0.04$ |
| 3B | Linear | α=2.9 | $0.77 \pm 0.01$ | $0.86 \pm 0.02$ | $0.80 \pm 0.02$ | $0.63 \pm 0.01$ | $0.92 \pm 0.04$ |
| 3B | Linear | α=3.0 | $0.77 \pm 0.01$ | $0.86 \pm 0.02$ | $0.81 \pm 0.02$ | $0.63 \pm 0.01$ | $0.91 \pm 0.04$ |
| 3B | Linear | α=3.1 | $0.76 \pm 0.01$ | $0.86 \pm 0.02$ | $0.82 \pm 0.02$ | $0.63 \pm 0.01$ | $0.90 \pm 0.04$ |
| 3B | Linear | α=3.2 | $0.76 \pm 0.01$ | $0.85 \pm 0.02$ | $0.83 \pm 0.02$ | $0.63 \pm 0.01$ | $0.88 \pm 0.04$ |
| 3B | Linear | α=3.5 | $0.76 \pm 0.01$ | $0.85 \pm 0.02$ | $0.85 \pm 0.01$ | $0.63 \pm 0.01$ | $0.85 \pm 0.04$ |
| 3B | Linear | α=4.0 | $0.74 \pm 0.01$ | $0.82 \pm 0.02$ | $0.87 \pm 0.01$ | $0.62 \pm 0.01$ | $0.77 \pm 0.03$ |
| 1B | Rank | k=1 | $0.58 \pm 0.01$ | $0.77 \pm 0.02$ | $0.64 \pm 0.02$ | $0.38 \pm 0.00$ | $0.98 \pm 0.04$ |
| 1B | Rank | k=5 | $0.73 \pm 0.01$ | $0.85 \pm 0.02$ | $0.74 \pm 0.01$ | $0.57 \pm 0.02$ | $0.98 \pm 0.04$ |
| 1B | Rank | k=20 | $0.85 \pm 0.02$ | $0.87 \pm 0.02$ | $0.80 \pm 0.01$ | $0.81 \pm 0.02$ | $0.95 \pm 0.05$ |
| 1B | Rank | k=50 | $0.75 \pm 0.01$ | $0.75 \pm 0.02$ | $0.63 \pm 0.02$ | $0.75 \pm 0.02$ | $0.92 \pm 0.04$ |
| 1B | Rank | k=100 | $0.81 \pm 0.02$ | $0.86 \pm 0.02$ | $0.85 \pm 0.01$ | $0.73 \pm 0.01$ | $0.87 \pm 0.05$ |
| 1B | Rank | k=200 | $0.75 \pm 0.01$ | $0.76 \pm 0.02$ | $0.67 \pm 0.02$ | $0.72 \pm 0.01$ | $0.88 \pm 0.04$ |
| 1B | Rank | k=500 | $0.74 \pm 0.01$ | $0.75 \pm 0.02$ | $0.72 \pm 0.02$ | $0.72 \pm 0.01$ | $0.79 \pm 0.04$ |
| 1B | Rank | k=1000 | $0.72 \pm 0.02$ | $0.73 \pm 0.02$ | $0.75 \pm 0.02$ | $0.72 \pm 0.01$ | $0.71 \pm 0.04$ |
| 3B | Rank | k=1 | $0.60 \pm 0.01$ | $0.84 \pm 0.02$ | $0.72 \pm 0.02$ | $0.38 \pm 0.00$ | $0.99 \pm 0.04$ |
| 3B | Rank | k=5 | $0.81 \pm 0.01$ | $0.89 \pm 0.02$ | $0.82 \pm 0.01$ | $0.69 \pm 0.02$ | $0.97 \pm 0.04$ |
| 3B | Rank | k=20 | $0.85 \pm 0.01$ | $0.89 \pm 0.02$ | $0.86 \pm 0.01$ | $0.77 \pm 0.02$ | $0.93 \pm 0.04$ |
| 3B | Rank | k=50 | $0.76 \pm 0.01$ | $0.77 \pm 0.02$ | $0.67 \pm 0.02$ | $0.74 \pm 0.01$ | $0.90 \pm 0.04$ |
| 3B | Rank | k=100 | $0.81 \pm 0.02$ | $0.85 \pm 0.03$ | $0.89 \pm 0.01$ | $0.73 \pm 0.01$ | $0.82 \pm 0.06$ |
| 3B | Rank | k=200 | $0.76 \pm 0.01$ | $0.78 \pm 0.02$ | $0.73 \pm 0.02$ | $0.73 \pm 0.01$ | $0.84 \pm 0.04$ |
| 3B | Rank | k=500 | $0.76 \pm 0.01$ | $0.77 \pm 0.02$ | $0.77 \pm 0.02$ | $0.72 \pm 0.01$ | $0.78 \pm 0.04$ |
| 3B | Rank | k=1000 | $0.74 \pm 0.02$ | $0.74 \pm 0.02$ | $0.78 \pm 0.02$ | $0.72 \pm 0.01$ | $0.70 \pm 0.04$ |

# D APPLICATION TO IMAGE GENERATION

In this section we detail the experimental setup used to assess the quality of generated images and additionally present (i) distributional statistics of image quality generated using our divergence decoding setup and (ii) a random sample of generated images for qualitative analysis.

## D.1 EXPERIMENTAL SETUP

Each image in our sample is generated using class conditional generation using the default generation parameters of (Esser et al., 2021) for their ImageNet checkpoint. We fine-tune the parameters of auto-regressive transformer in this model to arrive at checkpoints for $p$ and $q$ using a peak learning rate of 10% of that used in (Esser et al., 2021) training for 10 epochs each over the retain and forget sets. We then generate image samples following Eq. 3 where the adjustment from $p$ and $q$ is based solely on the output from the auto-regressive transformer.

## D.2 MEASURING IMAGE CONTENT AND QUALITY

Image content is measured using VQAScore (Lin et al., 2024). This approach requires access to the log probabilities of the multi-modal LLM (MLLM) used to assess quality, therefore these assessments rely on the GPT-4o-mini rather than newer models such as GPT-5-nano for which the log probabilities are not exposed. When measuring perceptual quality, we use a MLLM-as-a-judge in a pairwise comparison configuration (Chen et al., 2024). Since this setup does not require access to the log probabilities, we leverage several state of the art small MLLMs for this task: Gemini 2.5 Flash-Lite, GPT-5-nano, and Qwen3-VL 8B.

## D.3 DISTRIBUTIONAL PROPERTIES OF GENERATED IMAGES

Ideally, samples from the model would no longer exhibit image semantics associated with the data in the forget set $F$, while retaining high perceptual quality relative to the retain set $R$. Following prior work (e.g., Heusel et al., 2017), we measure the quality of the generated images using the Fréchet Inception Distance (FID).

We assess performance by computing the FID between three pairs of data: (i) baseline images from the retain set and generated images using classes from the retain set (FID($B_R,G_R$)), (ii) baseline images from the forget set and generated images using classes from the forget set (FID($B_F,G_F$)), and (iii) baseline images from the retain set and generated images using classes from the forget set (FID($B_R,G_F$)).

Efficacy in this setting preserves perceptual quality relative to the retain set, i.e., low FID($B_R,G_R$) and low FID($B_R,G_F$), while increasing the distance between the forget set and images generated based on those classes, i.e., high FID($B_F,G_F$). In Table 8, we present FID statistics for a variety of decoding setups. For the linear setup, an $\alpha = 1$ seems to work well, e.g., a roughly 33% increase in FID($B_F,G_F$) with only a 5% increase in FID($B_R,G_R$) relative to the baseline. In contrast, the topk based methods appear to require much larger values of $k$ to be effective.

Table 8: Content analysis of images generated using various divergence decoding setups.

| Method | Config | FID($B_R,G_R$) ↓ | FID($B_F,G_F$) ↑ | FID($B_R,G_F$) ↓ |
|---|---|---|---|---|
| Baseline | — | 18.2 | 18.0 | 30.1 |
| Linear | $\alpha = 1$ | 19.2 | 24.1 | 27.3 |
| Linear | $\alpha = 2$ | 20.5 | 28.7 | 26.8 |
| Linear | $\alpha = 5$ | 22.8 | 31.6 | 25.8 |
| Linear | $\alpha = 10$ | 22.6 | 31.4 | 25.3 |
| Rank | topk=20 | 19.1 | 20.0 | 29.2 |
| Rank | topk=100 | 20.1 | 22.1 | 28.7 |
| Rank | topk=250 | 21.1 | 28.1 | 26.0 |

## D.4   IMAGE GENERATION SAMPLES

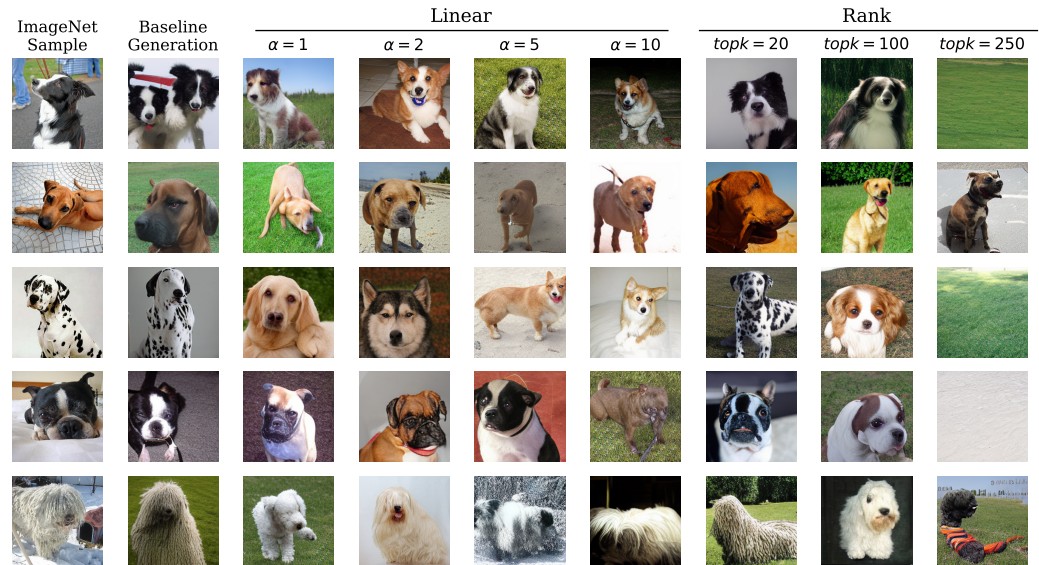

Figure 10: Random sample of image generations for classes in the forget set.

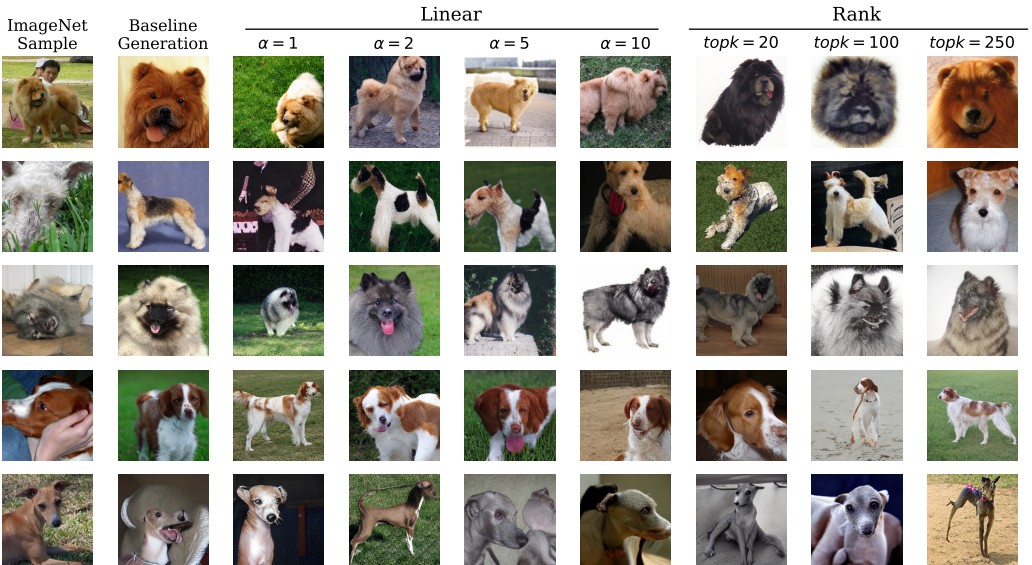

Figure 11: Random sample of image generations for classes in the retain set.

