# OpenReview forum: "Generalized Inference Time Unlearning --- Effective for A Fraction of the Cost"
_ICLR.cc/2026/Conference — Submitted to ICLR 2026_

### Official Review · Reviewer_CrGY · 2025-11-01

**Soundness:** 2
**Presentation:** 2
**Contribution:** 2
**Rating:** 4
**Confidence:** 4

**Summary:**

This paper proposed an inference-based unlearning method, reducing the overhead of fine-tuning the LLM. The method based on divergence decoding, which two distributions are built to guide the LLM token decoding. The experimental results shows its effectiveness.

**Strengths:**

1. The paper proposed time-based unlearning benchmark, enabling more diverse evaluation.

2. The proposed method is applicable to different data types, including text and image.

**Weaknesses:**

1. As pointed by the limitation 2 in the paper, as this method does not really change the internal representation of a model, it is more like guardrails instead of unlearning. Moreover, could the authors provide a concrete scenario how the proposed method would be used?

2. The inference time cost analysis miss the important factor: run time latency, as the proposed method will have to recompute the distribution every token, could the authors provide the total runtime before and after applying the proposed unlearning approach? Unlike other approaches, like GA, it could be costly during the fine-tuning but it should work as is during the inference.

**Questions:**

1. I wonder what is the connection of the proposed method to the LLM watermarking, which is also tiling the distribution in a certain way.

2. For time-based unlearning benchmark, what is the difference if we simply categorize those to-be-forgotten timeline into forget set and remaining go to retain set?

3. What is the effect of the ratio of model size of p and q (not absolute size)?

---

> ### Author Response · Authors · 2025-12-03
>
> ## Weaknesses
> ### Internal representations unchanged.
> We agree that the internal representations of P are not changed. We have updated the paper title and writing to reflect that the base model is not being changed but instead being guided.
>
> ### Concrete use case.
> Here are several use cases which motivated the work:
> 1. Compliance with “Right to be forgotten” regulation. These pervasive statutes under the EU’s GDPR and other jurisdictions such as California provide individuals with the right to request that platforms remove information about them, e.g., from search results. These requests are numerous and ad hoc which poses a challenge for traditional unlearning methods as highlighted in Figure 5. Our method, specifically the n-gram based setup, enables fast and effective unlearning of the content in these requests.
> 2. Addressing copyright violations. There are numerous examples of alleged copyright infringement by model providers. We envision a scenario where our method can be used to quickly eliminate such violations, particularly the most egregious cases where copyrighted material is regurgitated verbatim.
> 3. Financial backtesting. In many applications in finance, such as testing trading strategies or stress testing of banks, it is desirable to conduct a “backtest” which relies only on information which was available to decision makers at the time. Since LLMs are trained on long time-series of content, such backtests are not currently possible, e.g., frontier models are aware of the 2008 US financial crisis thereby making it difficult to realistically assess the performance of an LLM in a decision making role at this time.
>
> ### Run time latency.
> This is an excellent point, we have added an empirical analysis of more than 1,200 models to assess the run time latency and compute requirements of our method (see Figure 7 and 8). We find that in a realistic production environment run time latency typically increases by less than 0.1%, while compute scales similarly to our theoretical analyses. In our naive setups for MUSE and TOFU (unbatched, models running sequentially) we do vastly exceed these theoretical analyses
>
> ## Questions
> ### I wonder what is the connection of the proposed method to the LLM watermarking, which is also tiling the distribution in a certain way.
> This is an interesting observation since it also subtly adjusts the output of the model. Since we added a new section in the literature about inference-time steering, we included a sentence about this as well.
>
> ### For time-based unlearning benchmark, what is the difference if we simply categorize those to-be-forgotten timeline into forget set and remaining go to retain set?
> This is essentially what we were envisioning. However, we removed the time unlearning benchmark from the paper in order to focus on the method.
>
> ### What is the effect of the ratio of model size of p and q (not absolute size)?
> This is the right way to think about the sizing, we have updated the paper to reflect this (see, e.g., Figure 2). If your question referred to p and q not being the same size – our goal is to minimize divergence between p and q, or else it will lead to unstable outputs. If they are different sizes the logits would have a different peakiness and it would cause all kinds of issues.

---

### Official Review · Reviewer_UHuj · 2025-11-01

**Soundness:** 2
**Presentation:** 2
**Contribution:** 2
**Rating:** 4
**Confidence:** 3

**Summary:**

This paper introduces Divergence Decoding (DD), an inference-time technique for unlearning. The technique involves finetuning 2 smaller models: 1 on a "retain set" that includes knowledge that should not be forgotten, and 1 on the "forget" set contains information on the concept that should be forgotten. The logits of the base model are then adjusted by the difference in logits of the forget and retain models. The authors propose a linear method of DD, as well as a rank-based method. They also introduce a new time-unlearning benchmark

**Strengths:**

- The problem of Unlearning is important and timely
- Inference-time unlearning techniques, such as the proposed approach, are needed as finetuning is costly and prone to harming generalizability.
- The technique of using two proxy models to adjust the logts of the base model is clever
- The proposed approach is somewhat backed by theory, as the authors relate it to Product of Experts and importance sampling.

**Weaknesses:**

- The experimental results lack error bars / confidence intervals.
- Unclear whether DD outperforms NP on verbatim knowledge of the forget set on MUSE (figure 1)
- The biggest weakness in my eyes is the time-unlearning benchmark. I may be missing something, but frankly I do not see how this fits in with the rest of the paper. No unlearning methods, DD or otherwise, are evaluated on the proposed Time Unlearning dataset in the main paper. I also don't understand how this dataset relates to unlearning, since it focuses on lookahead bias rather than data removal per se. The paper would be much stronger if it removed this dataset and instead performed a more comprehensive analysis on standard unlearning benchmarks.

**Questions:**

- How does the Time Unlearn dataset relate to Unlearning? Why are no Unlearning methods evaluated on it?
- I don't understand Figure 3. Can you explain in more detail what each axis represents? Why is a flat line desirable? Is DD performing poorly compared to NPO and SimNPO?

---

> ### Author Response · Authors · 2025-12-03
> **Response**
>
> ## Weaknesses:
> ### The experimental results lack error bars / confidence intervals.
> We have added error bars throughout the paper wherever possible. For TOFU specifically, the original paper and benchmark code does not include error bars, and this is difficult to do since there are over 20 metrics which are nonlinearly aggregated to form the final score, which is why we originally did not provide it. All experimental results now provide 99% CI
>
> ### Unclear whether DD outperforms NP on verbatim knowledge of the forget set on MUSE (figure 1)
> We appreciate you highlighting this. We have added lines showing the efficient frontier of unlearning methods on a utility/unlearning tradeoff, demonstrating that DD performs at the frontier on verbatim and significantly outperforms the frontier on Q&A.
>
> ## Questions
> ### Time Unlearning Dataset.
> The original purpose of the time-unlearning dataset was to focus on what we think will be the next frontier of unlearning which will be unlearning large amounts of time information in order to allow LLMs to be used for applications such as backtesting in finance. Since we considered the dataset a validation set, and unlearning on all time based information is far outside the scope of our resources, we initially chose not to train on it. With that said, we decided to remove it (as discussed above) to better focus on the method itself.
>
> ### I don't understand Figure 3. Can you explain in more detail what each axis represents? Why is a flat line desirable? Is DD performing poorly compared to NPO and SimNPO?
> Thank you for this feedback, you are correct that it was unclear. We have redesigned the figure and the metric being measured to track drift from the baseline forget set. This shows the same story as the previous, which is that all of the methods have the same comparable degradation, except GradDiff, which has a very large degradation. This is consistent with results in the original MUSE paper.

---

### Official Review · Reviewer_tTAn · 2025-11-04

**Soundness:** 3
**Presentation:** 3
**Contribution:** 3
**Rating:** 4
**Confidence:** 3

**Summary:**

This paper introduces a new paradigm, namely Inference-Time Unlearning (ITU), which aims to remove unwanted knowledge generation from large language models without modifying their weights. The approach trains two much smaller auxiliary models — a forget expert fine-tuned on the forget set, and a retain expert fine-tuned on the retain set. During inference, these experts adjust the logits of the base model using a method called Divergence Decoding (DD), which steers the output distribution away from tokens upweighted by the forget expert and toward those upweighted by the retain expert. Two variants are proposed: a linear logit adjustment and a rank-based token suppression. This inference-time mechanism achieves effects that are similar to “unlearning” of targeted content while being orders of magnitude more computationally efficient than gradient-based methods and preserving general utility across MUSE and TOFU benchmarks.

**Strengths:**

- **New paradigm on LLM output steering.** The proposed method aims to efficiently approximate the data distribution of a target large model ($\hat Q$) using two introduced much smaller fine-tuned experts ($p,q$) plus the original model ($P$), achieving substantial computational savings while maintaining controllability.


- **The formulation is concise.** The derivation connecting Divergence Decoding to Product of Experts and importance sampling is mathematically coherent and provides interpretability.

- **Extensive empirical verification.** The author demonstrates the effectiveness of proposed methods on multiple benchmarks and visual distributions: MUSE, TOFU, image distribution and the introduced “Time-unlearning benchmark”

**Weaknesses:**

- **Dependence on auxiliary models.** The method assumes well-trained retain and forget experts but gives little detail on how to build them when data are limited or noisy, leaving room for bias or domain overlap artifacts.

- **Evaluation gaps in image generative domains.** The image unlearning experiment relies solely on FID scores, which are inadequate for assessing semantic forgetting. The work would be beneficial to employ automated analysis verifying that specific visual concepts are unlearned while others remain intact, or include visual examples for qualitative evaluation.

- **Conceptual ambiguity.** The method steers outputs but does not alter the model’s internal representations or parameters. Hence, it does not “remove” knowledge but suppresses its expression on targeted contents. This is better categorized as inference-time filtering or redirection, not unlearning in the formal sense.

- **Missing relevant work discussion.** Training-free steering-based approach for controlled model generation is not a completely new paradigm. While this work focuses on logit space steering, prior work on activation space steering [1][2] needs to be discussed, and potentially, compared efficiency, as steering-based methods all claim benefits on computational efficiency and utility preservation.

[1] Steering Language Models with Activation Engineering

[2] Programming Refusal with Conditional Activation Steering

**Questions:**

- **Clarification on experimental setup.** Are there any specific retain/forget set curation involved?

- Could you explicitly explain how proposed method handle non-targeted content generation?

- See weaknesses for other questions/suggestions

---

> ### Author Response · Authors · 2025-12-03
> **Response**
>
> ## Weaknesses:
> ### Dependence on auxiliary models. The method assumes well-trained retain and forget experts but gives little detail on how to build them when data are limited or noisy, leaving room for bias or domain overlap artifacts.
>
> Originally, we provided little detail on this because we found that training the models was quite simple, even in the context of noisy or limited data. For example:
>
> -	The n-gram based models, which require very little data to train, can be highly effective
> -	For training the transformer-based models, we found that using standard fine-tuning practices worked well
>
> We have now clarified this in the paper and want to note that existing unlearning methods typical involve tuning as well.
>
> ### Evaluation gaps in image generative domains.
> Thank you for this excellent point. Following your feedback, we have added several random samples of images to the paper and perform additional evaluations per the current state of the art. Specifically, we assess the content of the images using VQAScore (Lin et al, 2024) and the perceptual quality using pairwise comparisons via an MLLM-as-a-judge setup (Chen et al, 2024).
>
> ### Conceptual ambiguity.
> We have updated the writing in the paper to address this and highlight that the method is effectively steering the output of the target model. The way we thought about the method is that “inference time” generally refers to forward passes involving a fixed set of weights. Along these lines, the only way to do “unlearning” at inference time is through the addition of some mechanism which does not involve changing the weights of the target model directly.
>
> ### Missing relevant work discussion.
> Thank you for highlighting this, we have added this work to our related literature discussion.
>
> ## Questions:
> ### Clarification on experimental setup. Are there any specific retain/forget set curation involved?
> For the benchmarks, we use the specific retain and forget sets that are provided. As we discuss in section 4.1, we may or may not train the forget model on both retain and forget or just forget depending on the difference of the size of the sets, in order to minimize divergence between the different models. Regarding your point about the dependence on the auxiliary models, we found that balancing the relative sizes of the retain and forget set sizes was the only “new” factor which needed consideration—relative to typical fine-tuning—which needed consideration in order to arrive at good models. We found that when the retain set is much larger than the forget set, is is better to finetune $q$ on both the forget and retain set.
>
> ### Could you explicitly explain how proposed method handle non-targeted content generation?
> As we discuss in section 3.2, when $q \approx p$, there should be no change in the logits. Hence, during non-targeted content generation the differences in the models will be small and not affect the outputs. Empirically, we find this to be the case, i.e., there is effectively zero utility loss on the retain set.

---

### Public Comment · ~Vinith_Menon_Suriyakumar1 · 2025-11-18
**Missing a Highly Relevant Prior Work**

Dear Authors and Reviewers,

We want to surface our paper, UCD: Unlearning in LLMs via Contrastive Decoding, that was posted to ArXiv in June of this year, which also proposes inference-time unlearning.

Paper Link: https://arxiv.org/abs/2506.12097 (Posted on Thursday, June 12, 2025)

Similarly, our work proposes a new decoding approach that leverages two auxiliary smaller models which are then used to perform “contrastive decoding” and modify the distribution of the original model as close to the “retrained from scratch model”. We would encourage the authors to cite our work and also indicate the differences between their work and ours. Below, we list the differences and similarities that we see between the two works. **We emphasize that this work was posted four months before the ICLR submission deadline, giving ample time to be found and cited.**

Similarities:

- Motivation: Both our work and this submission propose a new type of unlearning paradigm that aims to address the cost and scalability issues of weight-based unlearning approaches for LLMs.
- Algorithm: Both works propose the use of two smaller auxiliary models, one trained on the forget set and one trained on the entirety of the data, to modify the logit distribution of the model we aim to unlearn on. The algorithm proposed in our work, UCD, subtracts the difference between the logits of the auxiliary models from the large model’s logits. This is exactly the same as the linear method proposed in the Divergence Decoding section of this work. Both papers build upon the contrastive decoding algorithm proposed in [1].
- Evaluations: Both works evaluate the performance of the algorithms on MUSE and TOFU.
- Results: Both works show that this algorithm can successfully perform unlearning on MUSE and TOFU. Often outperforming existing algorithms that perform unlearning via finetuning.

Differences

- Theoretical motivation for the proposed algorithm is different
- The subsequent analyses in both papers differ:
   - Our paper investigates whether bootstrapping with small unlearned models generated using existing unlearning techniques still leads to improvements
   - Our paper also shows that UCD scales to unlearning on Llama-70B using the same compute as finetuning-based algorithms
- This submission investigates the applicability of this algorithm beyond text and in time-based unlearning settings, such as backtesting

Overall, we would like to highlight that our work UCD preceded this submission and that the core algorithms and results on standard benchmarks in this paper are the same as our paper. Thus, the algorithm itself proposed is not a novel contribution in light of our work. Thus, we encourage the authors to highlight this in their paper, cite our work, and distinguish remaining differences.

[1] Li, Xiang Lisa, et al. "Contrastive decoding: Open-ended text generation as optimization." Proceedings of the 61st annual meeting of the association for computational linguistics (volume 1: Long papers). 2023.

---

> ### Author Response · Authors · 2025-12-03
>
> Thank you for bringing this to our attention. We disagree with your assertion and characterization of our work as derivative. An early version of our paper—which was a superset of your paper—was presented and/or discussed at our home university, AI labs, and several financial firms as early as August 2024. Consistent with this, our paper is substantially more developed, as detailed below.
>
> Considering official ICLR policy we are only expected to compare our work to papers which were published in a peer-reviewed venue on or after June 1, 2025, i.e., September deadline minus four months (https://iclr.cc/Conferences/2025/FAQ). Notably, this does not include your paper. That said, we recognize this is a fast-moving field and are happy to cite contemporaneous work, and now do so.
>
> Non-exhaustive list of differences:
>
> -	Extensive theoretical motivation for performing unlearning via the method. In contrast, Suriyakumar et al. have no theoretical motivation nor connection to prior theoretical work, i.e., product of experts and importance sampling, and simply propose it as an ad hoc method
> -	Detailed exploration of the utility vs. unlearning tradeoff, i.e., we aggregate these two metrics into a single “distance from the ideal retrained model” to capture the observed pareto optimal frontier of state of the art unlearning methods.
> -	Examination of all metrics in TOFU, based on the work of Dorna et. al (2025), using the score as the aggregation of over twenty different submetrics  in total, versus only considerings the  metrics of utility and forget quality.
> -	Exploration of rank-based application in addition to the linear approach. We show that this approach matches or exceeds the linear setup while being more robust to hyperparameter choice
> -	Implementation and evaluation of n-gram based models, which we show are extremely inexpensive to train and run inference on while providing frontier-level elimination of verbatim memorization
> -	Comprehensive ablation studies exploring hyperparameter choice, algorithm choice, model size choices
> -	Significantly more in-depth study of over- and under-unlearning, and privacy
> -	Analysis of the compute costs and runtime for more than 1,200 model configurations
> Preliminary exploration of generalizability outside LLMs vis a vis divergence decoding for latent diffusion models

---

### Author Response · Authors · 2025-12-03
**Meta-Response for AC**

We would like to thank the reviewers for their feedback and the AC for reviewing our paper. We have provided detailed responses to the feedback from each reviewer and below is a meta response for the AC covering the largest changes to the paper. Finally, we wish to note that the author of a related working paper—posted online in June and significantly less developed than our work—also commented on our submission. We disagree with his characterization of our paper as derivative and detail why in our response to his comment.

## Meta Response
### Concrete use cases.
Here are several use cases which motivated the work:
1. Compliance with “Right to be forgotten” regulation. These pervasive statutes under the EU’s GDPR and other jurisdictions such as California provide individuals with the right to request that platforms remove information about them, e.g., from search results. These requests are numerous and ad hoc which poses a challenge for traditional unlearning methods as highlighted in Figure 5. Our method, specifically the n-gram based setup, enables fast and effective unlearning of the content in these requests.
2. Addressing copyright violations. There are numerous examples of alleged copyright infringement by model providers. We envision a scenario where our method can be used to quickly eliminate such violations, particularly the most egregious cases where copyrighted material is regurgitated verbatim.
3. Financial backtesting. In many applications in finance, such as testing trading strategies or stress testing of banks, it is desirable to conduct a “backtest” which relies only on information which was available to decision makers at the time. Since LLMs are trained on long time-series of content, such backtests are not currently possible, e.g., frontier models are aware of the 2008 US financial crisis thereby making it difficult to realistically assess the performance of an LLM in a decision making role at this time.

### Dependence on auxiliary models. The method assumes well-trained retain and forget experts but gives little detail on how to build them when data are limited or noisy, leaving room for bias or domain overlap artifacts.
Originally, we provided little detail on this because we found that training the models was quite simple, even in the context of noisy or limited data. For example:

-	The n-gram based models, which require very little data to train, can be highly effective
-	For training the transformer-based models, we found that using standard fine-tuning practices worked well

We have now clarified this in the paper and want to note that existing unlearning methods typical involve tuning as well.

### Reorganized benchmarks and ablations.

We have reorganized the benchmarks section to present MUSE and TOFU together, incorporating substantial new ablation studies. Notably, Figure 2 now illustrates the impact of hyper-parameter and algorithm choices across MUSE memorization, Q&A tasks, and both the aggregate and privacy-excluded scores for TOFU.

### The experimental results lack error bars / confidence intervals.
We have added error bars throughout the paper wherever possible. For TOFU specifically, the original paper and benchmark code does not include error bars, and this is difficult to do since there are over 20 metrics which are nonlinearly aggregated to form the final score. We now implemented 99% CIs computed via hierarchical bootstrap resampling for TOFU.

### Run time latency.
This is an excellent point, we have added an empirical analysis of more than 1,200 models to assess the run time latency and compute requirements of our method (see Figure 7 and 8). We find that in a realistic production environment run time latency typically increases by less than 0.1%, while compute scales similarly to our theoretical analyses. In our naive setups for MUSE and TOFU (unbatched, models running sequentially) we do vastly exceed these theoretical analyses

### Evaluation gaps in image generative domains.
Following the reviewer's feedback, we have added several random samples of images to the paper and perform additional evaluations per the current state of the art. Specifically, we assess the content of the images using VQAScore (Lin et al, 2024) and the perceptual quality using pairwise comparisons via an MLLM-as-a-judge setup (Chen et al, 2024).

### Time Unlearning Dataset.
The original purpose of the time-unlearning dataset was to focus on what we think will be the next frontier of unlearning which will be unlearning large amounts of time information in order to allow LLMs to be used for applications such as backtesting in finance. Since we considered the dataset a validation set, and unlearning on all time-based information is far outside the scope of our resources, we initially chose not to train on it. With that said, we decided to remove it (as discussed above) to better focus on the method itself.

---

### Meta-Review · Area_Chair_4gkM · 2026-01-05

**Summary:**

1. The paper defines manipulating model outputs as "unlearning," a definition that drew disagreement from multiple reviewers. I also disagree, as this method retrains the model’s internal knowledge and remains vulnerable to attacks.
2. The proposed time unlearning dataset has limited relevance to the core problem, and the authors conduct no experiments using it.
3. Run time latency during inference.
4. Dependence on auxiliary models, which are challenging to train with limited data.
5. A public comment notes a similarity between this paper and an arXiv publication, which was posted four months before the submission deadline.

**Reviewer Concerns:**

Point 2 was addressed, and points 3 and 4 were partly addressed. But points 1 and 5 are still outstanding.

**Reviewer Scores:**

Reviewer UHuj expressed no concern regarding point 1 and may raise the score. I think the other reviewers will keep their scores unchanged.

---

### Decision · Program_Chairs · 2026-01-26

Reject